# Molecular Epidemiology of Porcine Circovirus Type 2 and Porcine Parvoviruses in Guangxi Autonomous Region, China

Pin Chen [1,2], Geng Wang [2], Jiping Chen [2], Weichao Zhang [2], Yin He [2] and Ping Qian [1,2,*]

[1] College of Veterinary Medicine, Huazhong Agricultural University, Wuhan 430070, China; chenpin@mail.hzau.edu.cn
[2] Guangxi Yangxiang Co., Ltd., Guigang 537100, China; 980278405@qq.com (G.W.)
[*] Correspondence: qianp@mail.hzau.edu.cn

**Abstract:** Both porcine circovirus (PCV) and porcine parvovirus (PPV) cause various diseases and bring huge economic losses to the global swine industry. PCV2 is associated with several diseases and syndromes, including postweaning multisystemic wasting syndrome (PMWS), porcine dermatitis and nephropathy syndrome (PDNS) and porcine respiratory disease complex (PRDC). The classical PPV is one of the most common causes of reproductive failure in pigs. In this study, tissue samples (tonsil, lung, mesenteric lymph node, hilar lymph node and superficial inguinal lymph node) were collected from pigs with suspected PCV2-associated disease (PCVAD), and viral DNA was extracted. The coinfection of PCV2 and PPV1–5 was detected using the polymerase chain reaction (PCR) method. Phylogenetic analysis based on capsid genes of PCV2, PPV2, PPV3 and PPV5 was conducted. The prevalence rates of PCV2, PPV1, PPV2, PPV3, PPV4 and PPV5 were 51.2%, 15.9%, 36.6%, 19.5%, 14.6% and 10.9% on the individual pig level, respectively. The coinfection rates of PCV2 with PPV1, PPV2, PPV3, PPV4 and PPV5 were 8.5%, 25.6%, 17.1%, 13.4% and 3.7%, respectively. The prevalence of PPV2, PPV3 and PPV4 in PCV2-positive pigs was significantly higher than those in PCV2-negative pigs. Phylogenetic analyses were performed using the neighbor-joining (NJ) method with 1000 bootstraps. The results indicated the existence of PCV2d and two major clusters of PPV2, PPV3 and PPV5 in the Guangxi Autonomous Region. PCV2d was the dominant strain, and the novel PPVs were circulating in domestic pigs in the Guangxi Autonomous Region. The results of this study underline the importance of active surveillance of PCV2d and PPVs from the swine population in this area.

**Keywords:** porcine circovirus type 2 (PCV2); porcine parvovirus (PPV); molecular epidemiology; coinfection; phylogenetic tree





## 1. Introduction

Increasing evidence suggests the high prevalence of PPVs in PCV2-infected cases. PPV1 has been shown to enhance lesions associated with PCV2 infection [1–3]. PPV2 and PPV4 also played important roles in PCVAD among infected pig [4–6]. The coinfection of PPVs with PCV2 can enhance PCVAD and PCV2-related lesions in pigs and interfere with successful vaccine protection [7,8]. Classical porcine parvovirus (PPV1) and novel porcine parvoviruses, designated porcine parvovirus 2 through 7 (PPV2-PPV7), are widespread in pig populations in Korea [9].

Porcine circoviruses including PCV1, PCV2, PCV3 and PCV4, which belong to the genus *Circovirus* in the *Circoviridae* family, are the smallest animal viruses, with a single-strand circular 1766–1768-nucleotides (nt)-long DNA genome [10,11]. PCV1 was first discovered from porcine kidney cell culture PK/15 (ATCC-CCL 33) [12]. About 8 years later, Tischer reported the PCV as a new, small, nonenveloped, icosahedral mammalian virus containing a circular single-stranded DNA genome, which is regarded as a contaminant of cells without pathogenicity. PCV1 infection is common in pigs, and pigs can produce antibodies against it [13]. In 1995, a "novel" PCV was isolated from diseased

pigs. These new viruses have been shown to be antigenically and genomically distinct from PCV1 isolates and have been designated as PCV2 viruses, which cause huge hazards to the swine industry worldwide and are associated with several diseases, such as post-weaning multisystemic wasting syndrome (PMWS), porcine dermatitis and nephropathy syndrome (PDNS), reproductive disorders and others, collectively named porcine circovirus-associated diseases (PCVAD). The PCV2 genome contains two major open reading frames (ORFs), and ORF2 encodes capsid (Cap) protein, which is an important target for designing next-generation vaccines and for the development of serological diagnostic methods against PCV2 [14,15]. PCV3 was first identified in domestic swine in 2016 in the USA. The virus detected in sows and aborted fetuses is associated with PDNS-like clinical signs and reproductive failure, which spreads widely throughout almost all pig and wild boar tissues in various countries, with a gradual increase in infection. PCV3 infection has been reported worldwide, and coinfection with other pathogens is also prevalent, but definitive proof of its pathogenicity is still lacking. In April 2019 in Hunan province, China, a new circovirus with a distinct relationship to other circoviruses was identified in several pigs with severe clinical disease syndromes, including respiratory and enteric signs as well as porcine dermatitis nephropathy syndrome (PDNS) [16]. Retrospective studies demonstrated that PCV4 has been present and circulating in swine herds for more than ten years. Several animals (mice, dogs and cattle) may serve as reservoirs for PCV, with the potential for cross-species transmission of PCV to swine. However, currently, it is not clear whether or not intermediate hosts are involved in interspecies transmission of PCV4 to swine herds [17].

Porcine parvoviruses (PPVs), which belong to the genus *Parvovirus* of the family *Parvoviridae*, are non-enveloped small single-stranded DNA viruses. Other members of the same genus include parvoviruses of cattle, cats, dogs, geese, mice, rats, rabbits, mink, chickens, geese and raccoons. The classical PPV1 is one of the most important causes of reproductive failure in pigs and was first isolated from dead and aborted fetuses in pig herds in 1965 in Munich, Germany, and it has had a worldwide distribution from then on [18]. PPV causes reproductive disorders in sows that can be summarized by the acronym SMEDI (stillbirth, mummification, embryonic death and infertility) [19]. It is a potential reproductive pathogen that causes clinical illness in adult swine or crosses the placental barrier to infect the porcine conceptus. PPV is one of the viruses that has been especially problematic worldwide.

Recently, several novel PPVs have been found in domestic pigs since the identification of PPV2 in sera from pigs in 2001 [20,21]. PPV2 was discovered by accident during a survey for the hepatitis E virus. The 5 kb genome was determined, and similarity with Muscovy duck parvovirus and bovine parvovirus 3 was established, but it did not cluster with any other known parvoviruses in the phylogenetic analysis. The emergence of PPV2 seemed to be detected in serum samples from pigs with "high fever"(a disease associated with porcine respiratory and reproductive syndrome virus, PRRSV) and PCV2-induced postweaning multisystemic wasting syndrome (PMWS). PPV3 was identified from porcine and bovine samples in Hong Kong and widely distributed in wild boars and domestic pigs [22]. PPV4 was initially discovered in pigs suspected of having porcine circovirus-associated disease (PCVAD) in North Carolina, and PPV4 was later reported in China [4,23]. Phylogenetically, the Chinese and American PPV4s are closely related and cluster in the same clade. They share greater than 99% nucleotide sequence identity in all three ORFs. The American PPV4 was first identified among swine suffering from an acute onset disease of high mortality in North Carolina, USA, during late 2005 [24]. PPV5 was identified from porcine lung samples, whereas PPV6 was identified from aborted pig fetuses. According to the existing criteria of parvovirus species classification, defined as <95% related by nonstructural gene DNA according to the International Committee on Taxonomy of Viruses, this newly isolated virus, with an identity of about 76% to the known sequences of PPV4, was considered a novel species and tentatively designated PPV5 [25]. Qiao described the identification and genome characterization of a novel parvovirus (PPV6) from pigs, the closest neighbors of which

were PPV4 and PPV5; it is approximately 6.1 kb in length, and the genomic organization of PPV6 is similar to PPV5 but not to PPV4, which contains an ORF3 in the middle of the viral genome [26]. Recently, six new genotypes of PPVs (PPV2 through PPV7) have also been detected in Chinese swine herds [27]. PPV6 DNA was identified in serum samples collected from domestic pigs in Poland [28]. PPV7, the most recently discovered PPV genotype, was first reported in US pigs in 2016 [29]. New amino acid substitutions have been observed in strains from several countries. Hot spots were found to be located on the capsid surface, and a surface profile distinct from the vaccine strains was observed [30,31].

This study investigated the prevalence of PCV2 and PPV1–5, as well as the coinfection in tissue samples recently collected from suspected PCVAD domestic pigs in the Guangxi Autonomous Region, China.

## 2. Materials and Methods

### 2.1. Collection of Samples

Five different kinds of tissue samples (tonsil, lung, mesenteric lymph node, hilar lymph node and superficial inguinal lymph node) were collected from 82 suspected PCVAD domestic pigs (among 15–112 days old) and stored at −80 °C; clinical signs include wasting with progressive weight loss, lethargy, dark-colored diarrhea, lymphadenopathy, and paleness or jaundice. Another 50 cord blood samples and 60 tonsil samples were randomly collected from sows. All the pigs had been farmed in different areas in the Guangxi Autonomous Region.

### 2.2. DNA Extraction

About 0.1 g of tissue sample was minced in Dulbecco's Modified Eagle Medium (DMEM) and homogenized using a Stomacher® 80 (Seward Laboratory Systems, Inc., Bohemia, NY, USA). Sample homogenates were centrifuged at $3000 \times g$ for 5 min, and supernatant aliquots were stored at −80 °C. Viral DNA was extracted from the supernatant aliquots or cord blood samples using a TIANamp Genomic DNA kit (Tiangen, Beijing, China) according to the instructions of the manufacturer.

### 2.3. Detection of PCV2 and PPVs in Various Samples Using PCR

The primers used to test for the presence of PCV2 and PPV1–5 are listed in Table 1, and the amplification product sizes are 326, 445, 338, 392, 406 and 479 bp, respectively. To determine the prevalence of PCV2 and PPVs, PCR was performed using specific primers with Premix Taq™ (TaKaRa, Tokyo, Japan). The primers used to detect PCV2, PPV1, PPV2, PPV3 and PPV4 were previously described [32,33]. The primer of PPV5 was designed using Primer Premier 5.0 software based on the conserved regions in the capsid gene of PPV5 isolate MI216 (JX896318). The PCR cycles consisted of an initial denaturation step at 95 °C for 5 min, and 30 cycles of 94 °C for 30 s, 58 °C (PPV1 and PPV5), 55 °C (PCV2, PPV2 and PPV3), 54 °C (PPV4) for 30 s, and 72 °C for 1 min, and a final extension at 72 °C for 10 min. PCR products were controlled using agarose gel electrophoresis.

### 2.4. Sequencing and Phylogenetic Analyses

MEGA 6.0 software was used for assembly, alignment and analysis of the genetic changes among different strains of PCV2 (7 from tonsil, 5 from superficial inguinal lymph node, 3 from hilar lymph node), PPV2 (4 from tonsil, 2 from lung), PPV3 (3 from tonsil, 1 from mesenteric lymph node) and PPV5 (2 from tonsil) full-length major capsid genes, which were amplified with special primers (Table 1). PCR was performed with LA Taq® (TaKaRa, Dalian, China) using the following thermal profile: 95 °C for 5 min, and 35 cycles of 94 °C for 30 s, 58 °C for 30 s, 72 °C for 1 min (PCV2), 2 min (PPV3), and 3 min (PPV2, PPV5), and a final extension at 72 °C for 10 min. PCR products were purified using a TIANgel Midi Purification Kit (Tiangen, Beijing, China) and then ligated directly into a TA cloning vector system (Promega, Madison, WI, USA) and sequenced using TSINGKE

Biological Technology (Wuhan, China). Multiple sequence alignments and phylogenetic analyses were performed using the neighbor-joining (NJ) method with 1000 bootstraps.

**Table 1.** Primers used for detection of PCV2 and PPVs and amplification of the major capsid genes of PCV2, PPV2, PPV3, and PPV5.

| Primer Name | Primer Sequence (5'–3') | Size of PCR Product (bp) |
|---|---|---|
| PCV2-DF [a] | ATGGCATCTTCAACACCCG | 326 |
| PCV2-DR [a] | GATTGTATGGCGGGAGGAGT | |
| PPV1-DF [b] | TGGTCTCCTTCTGTGGTAGG | 445 |
| PPV1-DR [b] | CAGAATCAGCAACCTCAC | |
| PPV2-DF [c] | TACCAGACAGGCGACAACAAT | 338 |
| PPV2-DR [c] | CAGGCATAGGAGGAATGAAGG | |
| PPV3-DF [d] | GCAGTCTGCGCTTAACTT | 392 |
| PPV3-DR [d] | CTGCTTCATCCACTGGTC | |
| PPV4-DF [e] | GGAGACACAGTAGTTAGGACCCC | 406 |
| PPV4-DR [e] | TTTTCATACCCCAAATGGATAG | |
| PPV5-DF [f] | TGGTCACCGAGAACAGAAAG | 479 |
| PPV5-DR [f] | ATGCACCTGGTAAGGATGTC | |
| PCV2-AF [g] | ACTTACAGCGCACTTCTTTC | 771 |
| PCV2-AR [g] | GAGTCTTTTTTATCACTTCG | |
| PPV2-AF [h] | ATGAGCGCTGCCGACGCGTGGAAG | 3099 |
| PPV2-AR [h] | TTATACACGATGAGCGCGTCCCTCTGG | |
| PPV3-AF [i] | ATGGCTGCGCCGGTAACG | 1668 |
| PPV3-AR [i] | TTACAATTTGCGGGAGAAATTC | |
| PPV5-AF [j] | ATGAGCTTTAGTGGGTATTC | 2976 |
| PPV5-AR [j] | TTATCTTCTCGCTCTAACAC | |

a. Primers used to the detect PCV2 in the tissue samples; b~f. Primers used to the detect PPV1, PPV2, PPV3, PPV4 and PPV5 in the tissue samples; g. Primers used to the clone the capsid gene of PCV2; h. Primers used to the clone the capsid gene of PPV2; i. Primers used to the clone the capsid gene of PPV3; j. Primers used to the clone the capsid gene of PPV5.

## 3. Results

### 3.1. Prevalence of PCV2 and PPVs on the Tissue Level

All of the primers used in this study are listed in Table 1. In order to observe the prevalence of PCV2 and PPVs in domestic pigs, samples were collected from 82 suspected PCVAD pigs (15–112 days old) and detected using the PCR method with specific primers. Among 520 tissue and cord blood samples, the prevalence rates of PCV2, PPV1, PPV2, PPV3, PPV4 and PPV5 were 47.5% (247/520), 7.7% (40/520), 22.1% (115/520), 14.8% (77/520), 2.9%(15/520) and 8.3% (43/520), respectively. As shown in Table 2, PCV2 existed in various tissues with a high prevalence rate. PPV1–5 also existed in all kinds of tissues, except cord blood samples in sows. The positive rates of PPV2 and PPV3 were significantly higher than those of PPV1, PPV4 and PPV5.

**Table 2.** Prevalence rates of PCV2 and PPVs on the tissue level.

| Age | Tissue | PCV2 | PPV1 | PPV2 | PPV3 | PPV4 | PPV5 |
|---|---|---|---|---|---|---|---|
| Sow | Cord blood | 22/50 | 0/50 | 0/50 | 0/50 | 0/50 | 0/50 |
| | Tonsil | 46/60 | 16/60 | 24/60 | 25/60 | 0/60 | 27/60 |
| Pig from 15 to 112 days | Tonsil | 31/82 | 2/82 | 23/82 | 8/82 | 2/82 | 2/82 |
| | Lung | 35/82 | 6/82 | 20/82 | 14/82 | 0/82 | 5/82 |
| | Mesenteric lymph node | 39/82 | 5/82 | 15/82 | 11/82 | 6/82 | 4/82 |
| | Hilar lymph node | 38/82 | 5/82 | 19/82 | 9/82 | 2/82 | 2/82 |
| | Superficial inguinal lymph node | 36/82 | 6/82 | 14/82 | 10/82 | 5/82 | 3/82 |
| | Total | 247/520 | 40/520 | 115/520 | 77/520 | 15/520 | 43/520 |

### 3.2. Prevalence of PCV2 and PPVs on the Individual Pig Level

The prevalence rates of PCV2 and PPV1–5 on the individual pig level were further analyzed among 82 suspected PCVAD pigs. As shown in Table 3, the prevalence rates of PCV2, PPV1, PPV2, PPV3, PPV4 and PPV5 were 51.2% (42/82), 15.9% (13/82), 36.6% (30/82), 19.5% (16/82), 14.6% (12/82) and 10.9% (9/82), respectively. The prevalence of PCV2 was high in pigs of different ages, except those that were 29–42 days old. Particularly, pigs aged 15–21 and 85–112 days old showed a significantly more positive prevalence rate of over 94%. The prevalence of PPV2 was higher than the other four PPVs.

**Table 3.** Prevalence rates of PCV2 and PPVs on the individual pig level.

| Age (Days) | PCV2 | PPV1 | PPV2 | PPV3 | PPV4 | PPV5 |
|---|---|---|---|---|---|---|
| 15–21 | 3/3 | 1/3 | 0/3 | 0/3 | 0/3 | 0/3 |
| 22–28 | 3/5 | 0/5 | 3/5 | 2/5 | 1/5 | 1/5 |
| 29–35 | 0/6 | 0/6 | 1/6 | 0/6 | 1/6 | 1/6 |
| 36–42 | 0/6 | 0/6 | 1/6 | 0/6 | 0/6 | 0/6 |
| 43–49 | 3/8 | 4/8 | 3/8 | 1/8 | 0/8 | 0/8 |
| 50–56 | 5/8 | 1/8 | 2/8 | 0/8 | 0/8 | 1/8 |
| 57–63 | 5/11 | 0/11 | 5/11 | 1/11 | 3/11 | 3/11 |
| 64–70 | 3/7 | 2/7 | 5/7 | 3/7 | 0/7 | 1/7 |
| 71–77 | 2/6 | 2/6 | 3/6 | 0/6 | 0/6 | 0/6 |
| 78–84 | 2/5 | 1/5 | 2/5 | 1/5 | 0/5 | 1/5 |
| 85–91 | 4/4 | 0/4 | 2/4 | 2/4 | 1/4 | 0/5 |
| 92–98 | 5/6 | 0/6 | 1/6 | 1/6 | 2/6 | 1/6 |
| 99–105 | 5/5 | 2/5 | 1/5 | 3/5 | 2/5 | 0/5 |
| 106–112 | 2/2 | 0/2 | 1/2 | 2/2 | 2/2 | 0/2 |
| Total | 42/82 | 13/82 | 30/82 | 16/82 | 12/82 | 9/82 |

### 3.3. Prevalence Rates of Double Infection on the Individual Pig Level

In order to observe the relationship between PPVs and PCV2, the prevalence rates of double infection among 82 suspected PCVAD pigs were conducted on the individual pig level. As shown in Table 4, the concurrent infection rates of PCV2 with PPV1, PPV2, PPV3, PPV4 and PPV5 were 8.5% (7/82), 25.6% (21/82), 17.1% (14/82), 13.4% (11/82) and 3.7% (3/82), respectively. The prevalence rates of PPV1–5 in PCV2-positive pigs were 16.7% (7/42), 50% (21/42), 33.3% (14/42), 26.2% (11/42) and 7.2% (3/42). By contrast, the prevalence rates of PPV1–5 in PCV2-negative pigs were 15% (6/40), 22.5% (9/40), 5% (2/40), 2.5% (1/40) and 15% (6/40). Moreover, $\chi^2$ test analysis showed significant associations between PCV2 and PPV2 ($\chi^2 = 6.678$, $p < 0.02$), PCV2 and PPV3 ($\chi^2 = 10.473$, $p < 0.01$), PCV2 and PPV4 ($\chi^2 = 9.204$, $p < 0.01$), and PPV1 and PPV2 ($\chi^2 = 5.523$, $p < 0.02$). The results indicated that the occurrence of these four pairs of viruses was not random in this study, and the double infection among PCV2 and PPV1, PPV2, PPV3, PPV4 and PPV5 is a common phenomenon in this area.

### 3.4. Genetic Diversity of PCV2 and Novel PPVs
#### 3.4.1. PCV2

In this study, the complete ORF2 encoding for capsid protein of 15 PCV2-positive tissue samples was identified and compared with 24 other PCV2 strains available in the GenBank database, which represent genotypes 2a, 2b, 2c and 2d. Phylogenetic analysis showed that all of the 15 sequences identified in this study were grouped with the recently described PCV2d discovered in China (HM038017, KC515029, KM272212, and JX948776) and Korea (KJ437506) (Figure 1a). The results of multiple alignments showed that the nucleotide homology of PCV2 ORF2 ranged from 99.7% to 100% with PCV2 strain BDH (HM038017) (Figure 1b). Comparison of the amino acid sequences of the capsid protein showed that nine strains are identical with PCV2 strain BDH, and the other six strains have one change in the same position (169), two shifted R to S, and four shifted R to T (Figure 1c).

**Table 4.** Prevalence rates of concurrent PCV2 and PPVs infection.

| Double Infection | | Number of Pigs | | | | C² Value | p-Value |
|---|---|---|---|---|---|---|---|
| | | +/+ | +/− | −/+ | −/− | | |
| PCV2 | PPV1 | 7 | 35 | 6 | 34 | 0.043 | 0.836 |
| | PPV2 | 21 | 21 | 9 | 31 | 6.678 | 0.01 * |
| | PPV3 | 14 | 28 | 2 | 38 | 10.473 | 0.001 * |
| | PPV4 | 11 | 31 | 1 | 39 | 9.204 | 0.002 * |
| | PPV5 | 3 | 39 | 6 | 34 | 1.294 | 0.255 |
| PPV1 | PPV2 | 9 | 4 | 21 | 48 | 5.523 | 0.019 * |
| | PPV3 | 1 | 12 | 15 | 54 | 0.625 | 0.429 |
| | PPV4 | 0 | 13 | 12 | 57 | 1.439 | 0.230 |
| | PPV5 | 3 | 10 | 6 | 63 | 1.078 | 0.299 |
| PPV2 | PPV3 | 7 | 23 | 9 | 43 | 0.440 | 0.507 |
| | PPV4 | 4 | 26 | 8 | 44 | 0.000 | 1.000 |
| | PPV5 | 6 | 24 | 3 | 49 | 2.621 | 0.105 |
| PPV3 | PPV4 | 4 | 12 | 8 | 58 | 0.834 | 0.361 |
| | PPV5 | 1 | 15 | 8 | 58 | 0.052 | 0.819 |
| PPV4 | PPV5 | 1 | 11 | 8 | 62 | 0.000 | 1.000 |

*, significant ($p < 0.05$).

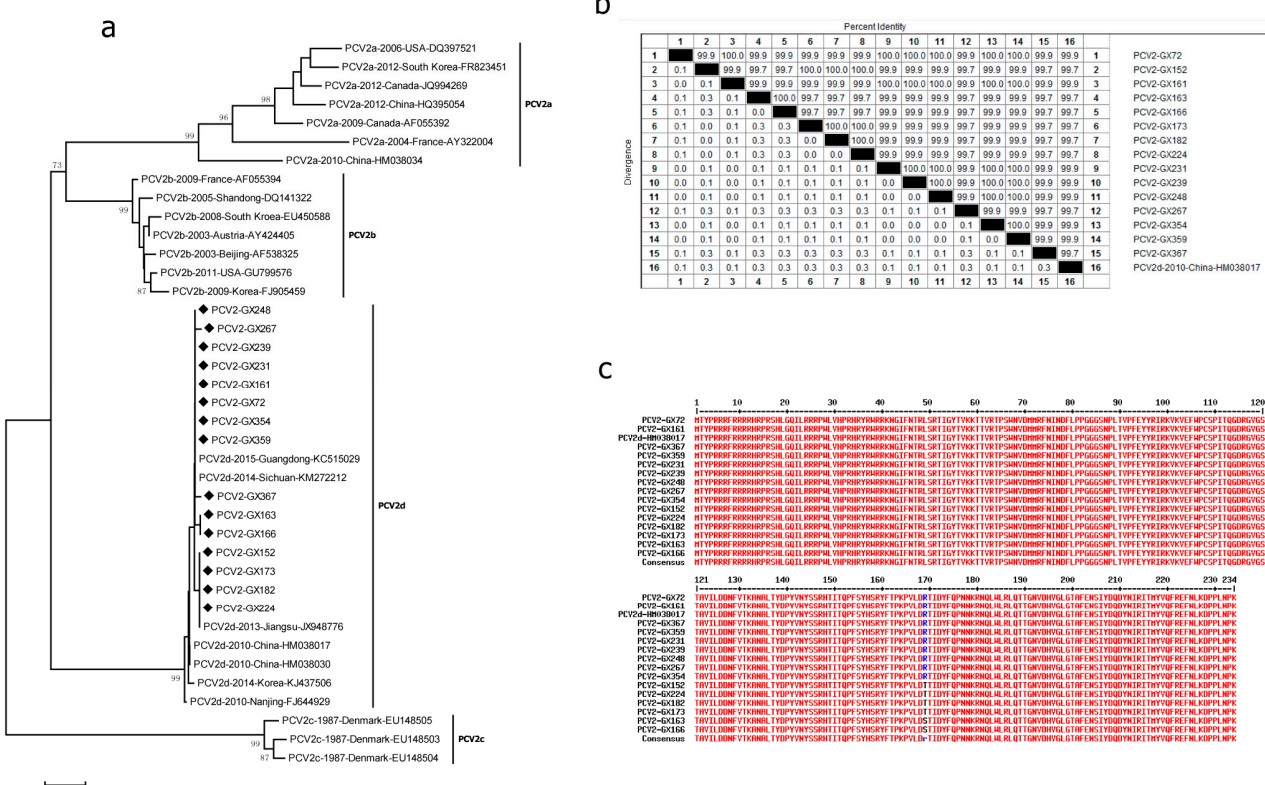

**Figure 1.** Genetic diversity of PCV2. Phylogenetic trees were constructed using the neighbor-joining method based on the complete major capsid genes of PCV2 strains archived in GenBank, with 1000 bootstrap replicates, using MEGA6.0 software. The nucleotide homology and amino acid difference were also conducted. Homology comparison was constructed using the Clustal W method, using DNAStar software. Comparison of the amino acid was constructed using online software (http://multalin.toulouse.inra.fr/multalin/ (accessed on 28 Jan 1998)). (**a**) Phylogenetic tree based on the complete major capsid genes of PCV2; (**b**) homology comparison of the complete capsid genes of PCV2; (**c**) comparison of the amino acid of our PCV2 capsid proteins with the respective reference strain BDH (HM038017). Newly identified PCV2 strains in this study are indicated with "◆". Reference sequences retrieved from GenBank are indicated by their years of isolation, origin and accession numbers.

3.4.2. PPV2

Six amplified PPV2 ORF2 sequences (PPV2-GX2, PPV2-GX27, PPV2-GX31, PPV2-GX200, PPV2-GX-Y3 and PPV2-GX103 strains) from PPV2-positive samples in this study were compared with nine other PPV2 strains available in the GenBank database. All fifteen sequences were roughly divided into two clusters, tentatively called cluster A and cluster B (Figure 2a). The sequences of PPV2-GX2, PPV2-GX27, PPV2-GX31 and PPV2-GX200 strains belonged to cluster A together with the Chinese strains (GU938300 and GU938301), Romania strains (KC701310 and KC701312), and USA strain (JX101461). The other two sequences of PPV2-GX-Y3 and PPV2-GX103 strains belonged to cluster B together with the Romania strains (JQ860246 and JQ860248), Myanmar strain (AB076669) and USA strain (KF725662). All 15 sequences were found to display a high nucleotide homology (93.1–99.8%) (Figure 2b). Analysis of the amino acid sequences showed that the amino acid difference in six strains from PPV2-positive samples ranged from 0.5% to 5.2%.

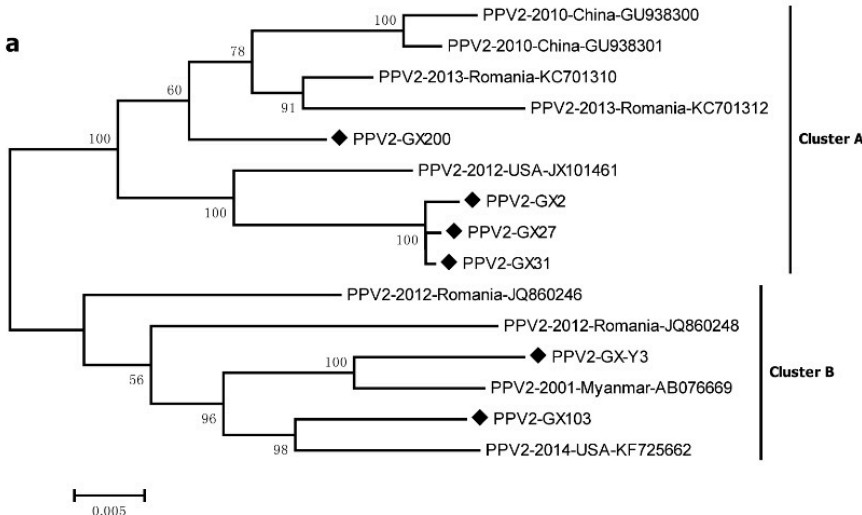

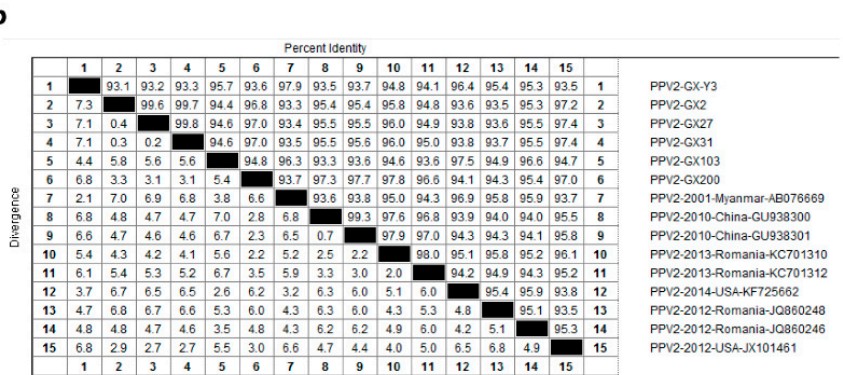

**Figure 2.** Genetic diversity of PPV2. (**a**) Phylogenetic tree based on the complete major capsid genes of PPV2; (**b**) homology comparison of the complete capsid genes of PPV2. Newly identified PPV2 strains in this study are indicated with "◆". Reference sequences retrieved from GenBank are indicated by their years of isolation, origin and accession numbers.

3.4.3. PPV3

For PPV3, the capsid genes of four PPV3-positive samples (GX162, GX352, GX357 and GX-Y6 strains) were sequenced and compared with other PPV3 reference strains in Hong Kong and Romania. The 19 sequences were divided into two main clusters. All of our four sequences belonged to cluster A and were grouped with Hong Kong strains (EU200672–EU200677) and Romania strains (JF738357, JF738364, and JQ868702) (Figure 3a). The maximum nucleotide difference in the four strains was 1.1%, whereas that of all

the 19 PPV3 strains was 2.4% (Figure 3b). Further analysis based on the amino acid sequences revealed that the maximum amino acid difference in our four strains was only 1.3%. Compared with the PHoV strain HK3 (EU200673), strain GX162 demonstrated only two changes (V56T and V252A), strain GX352 displayed three changes (V56T, V252A and A486T), strain GX357 showed three changes (V56T, V252A and R257G), and strain GX-Y6 exhibited five changes (I13V, V56T, S114P, S239P and P524L) (Figure 3c).

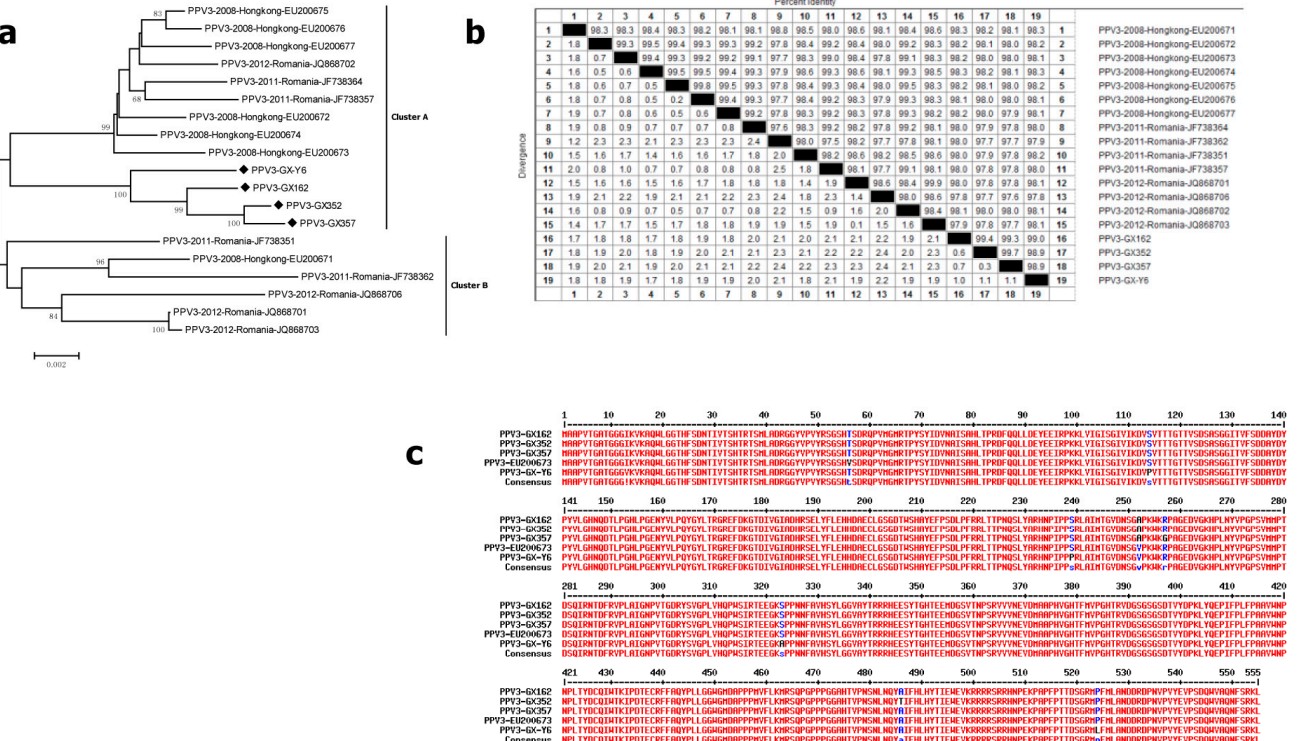

**Figure 3.** Genetic diversity of PPV3. (**a**) Phylogenetic tree based on the complete major capsid genes of PPV3; (**b**) homology comparison of the complete capsid genes of PPV3; (**c**) comparison of the amino acid of our PPV3 capsid proteins with the respective reference strain HK3 (EU200673). Newly identified PPV3 strains in this study are indicated with "◆". Reference sequences retrieved from GenBank are indicated by their years of isolation, origin and accession numbers.

### 3.4.4. PPV5

The capsid genes of two PPV5-positive samples (PPV5-GX-Y3 and PPV5-GX159) were compared with six other PPV5 reference strains. All eight sequences were roughly divided into two clusters, and the sequences of PPV5-GX-Y3 and PPV5-GX159 belonged to cluster A and were grouped with the USA strains (JX896318–JX896321) (Figure 4a). The nucleotide homology of all eight sequences ranged from 98.6% to 100% (Figure 4b), and the maximum amino acid difference in all eight strains was only 1.3%. These findings indicated that PPV5 also existed in Guangxi, and the strains detected from this area exhibited a high homology with the USA strains.

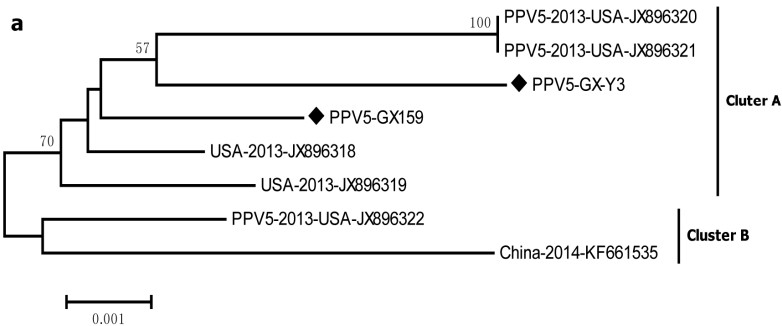

**Figure 4.** Genetic diversity of PPV5. (**a**) Phylogenetic tree based on the complete major capsid genes of PPV5; (**b**) homology comparison of the complete capsid genes of PPV5. Newly identified PPV5 strains in this study are indicated with "◆". Reference sequences retrieved from GenBank are indicated by their years of isolation, origin and accession numbers.

## 4. Discussion

PCV2 is one of the most significant pathogens causing large economic losses annually to the global swine industry. In this study, the prevalence rate of PCV2 was 47.5% on the tissue level, examining 520 tissue samples collected from 82 suspected PCVAD domestic pigs in the Guangxi Autonomous Region. Meanwhile, the positive rate of PCV2 was high in various tissue samples, including tonsils, cord blood, lungs, mesenteric lymph nodes, hilar lymph nodes and superficial inguinal lymph nodes (Table 2). The prevalence rate of PCV2 in the tonsil samples (76.7%, 46/60) was higher than that in cord blood (44%, 22/50) collected from sows (Table 2). Furthermore, a high prevalence of PCV2 was detected in different ages, except for those 29–42 days old. In particular, pigs aged 85–112 days old showed a >80% positive rate on the individual pig level (Table 4).

Moreover, the prevalence of PPV1–5 was also detected, and the prevalence rates of PPV1, PPV2, PPV3, PPV4 and PPV5 were 7.7%, 22.1%, 14.8%, 2.9% and 8.3% on the tissues level, respectively. PPV1–5 could be detected in all kinds of tissues, except the cord blood, which may be related to PPVs failing to cross the placental barrier. The positive rates of PPV2 and PPV3 were significantly higher than those of PPV1, PPV4 and PPV5. The prevalence rates of PPV1, PPV2, PPV3 and PPV5 in tonsil and lung samples were much higher than those in other tissue samples. Meanwhile, PPV2 was the predominant PPV type in those pig farms.

Coinfections of different PCV2 genotypes, as well as with other swine pathogens, may contribute to the development of more severe clinical symptoms in pigs [34–38]. Several reports showed that a high prevalence of PPVs was detected in PCV2-infected cases, and the coinfection of PPVs with PCV2 could enhance lesions associated with PCV2 infection [1,2,5,6]. Our study showed that the coinfection rates of PCV2 with PPV1, PPV2, PPV3, PPV4 and PPV5 were 8.5%, 25.6%, 17.1%, 13.4% and 3.7%, respectively. These results indicate that coinfections between PCV2 and PPVs were common in pig herds. Compared with the data surveyed by Sun JH et al., the positive rate of PCV2 was similar to their data (47.5% vs. 47.33%). Nonetheless, the positive rate of PPV1 was a little higher (7.7% vs.

5.56%) and the positive rates of PPV2, PPV3 and PPV4 were significantly lower in this study (22.1% vs. 39.56% for PPV2, 14.8% vs. 45.11% for PPV3, and 2.9% vs. 21.56% for PPV4) [8]. The PPV5 positive rate was significantly higher than that of the North American strain (8.3% vs. 3.4%) [7]. Analyses of double infection among PCV2 and PPVs also showed that the prevalence of PPV1–4 in PCV2-positive pigs was significantly higher than that in PCV2-negative pigs. Interestingly, we also observed the coinfection of PPV1–5 in this study.

There are five genotypes of PCV2, including PCV2a, PCV2b, PCV2c, PCV2d and PCV2e [38–41]. PCV2a was the predominant genotype until the emergence of PCV2b in 2005 [24]. PCV2c has been identified in Danish archive tissue samples and in a fetal pig [42,43]. PCV2d was isolated from the case of PCV2 vaccine failures in the USA and South Korea and became the prevalent strain in recent years [15,44–46]. Though PCV2a, PCV2b, PCV2d and PCV2e are identified in different areas in China, the phylogenetic analysis based on the sequences of ORF2 showed that all 15 PCV2-positive tissue samples were divided into genotypes PCV2d in the current study (Figure 1a). Our data are in agreement with results of a previous study showing that PCV2d isolates became predominant and widely distributed in pig farms isolated since 2009 [47]. There have been many reports on coinfections of different PCV2 genotypes in the field [32,40,48]. Previous research also showed that PCV2a was only detected in non-vaccinated pigs; the prevalence of the dominant epidemic PCV2d may be related to the widespread PCV2 vaccination in the Guangxi Autonomous Region [49].

Comparative phylogeny and evolution of novel PPVs based on the major capsid protein showed that the high rate of evolution of porcine parvoviruses was similar to that of RNA viruses [50]. In order to investigate the genetic changes among the different novel PPV strains, the full-length capsid genes of PPV2, PPV3 and PPV5 were further sequenced, and phylogenetic analysis was conducted. PPV2-GX2, PPV2-GX27, PPV2-GX31 and PPV2-GX200 strains joined to cluster A, and the other two strains joined to cluster B. There was a high nucleotide homology (93.1–99.8%) of all six strains with reference PPV2 strains (Figure 2). Meanwhile, all of our four PPV3 sequences in this study belonged to cluster A and were grouped with Hong Kong strains and Romania strains (Figure 3), which is consistent with the previous study [8]. Phylogenetic trees indicate that two PPV5 strains belonged to cluster A and grouped with the USA strains but were not clustered with the Chinese strain. Our study showed that multiple porcine parvoviruses infected pigs and provided more molecular data on novel PPV strains in China, necessitating close surveillance of PPVs in this region.

## 5. Conclusions

In summary, we report the prevalence of economically important PCV2 and PPVs and their coinfection in domestic pigs farmed in the Guangxi Autonomous Region. Although the PCV2d strains have similar antigenic reactivity and the commercial PCV2a vaccines could provide cross-protection against PCV2d challenge to a certain extent, the surveillance of predominant PCV2d and PPVs in farms will contribute to understanding the epidemiology of these diseases and designing novel vaccines to offer effective control measures in this region.

**Author Contributions:** P.C.: sampling, data collection, molecular and statistical analyses, manuscript writing. G.W.: sampling, statistical analyses, manuscript writing. J.C.: data analyses. W.Z.: manuscript reviewing, interpretation. Y.H.: data analyses. P.Q.: study design, supervision, manuscript reviewing. All authors have read and agreed to the published version of the manuscript.

**Funding:** This study was supported by Yingzi Tech and Huazhong Agricultural University Intelligent Research Institute of Food Health (IRIFH 202301) and the Research and Technological Development project of Guangxi Autonomous Region, China (14125008-2-18).

**Institutional Review Board Statement:** Not applicable.

**Informed Consent Statement:** Not applicable.

**Data Availability Statement:** Data developed in this study will be made available on request to the corresponding authors.

**Acknowledgments:** The authors would like to thank all participants. We are most grateful to Yangxiang Group (Guigang, China) for kindly providing us with all of the clinical tissue samples from suspected PCVAD domestic pigs.

**Conflicts of Interest:** The authors declare no competing interests.

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
