# Peer review of "Molecular Epidemiology of Porcine Circovirus Type 2 and Porcine Parvoviruses in Guangxi Autonomous Region, China"

_2036-7481, doi:10.3390/microbiolres14030090_

Round 1

Reviewer 1 Report

The authors report the prevalence of PCV2 and PPVs in this manuscript with convincing data and analysis, emphasizing the prevalence and huge threat of co-infection of PCV2 with different types of PPV. The manuscript is overall acceptable contingent upon minor modification.  

1.       Introduction paragraph 2: “in North Carolina and USA”

2.       Writing style of amino acid mutation should be “V56T”, “V252A”, “A486T”

3.       Figure 3 legend: “(c).” but not “c:”

4.       Format of the references is not unified

Author Response

Author's Reply to the Review Report (Reviewer 1)

1.Introduction paragraph 2: “in North Carolina and USA”

The author had modified the wrong expression as “in the North Carolina”.

2.Writing style of amino acid mutation should be “V56T”, “V252A”, “A486T”

The style of amino acid mutation had been modified by the authors according to the general rules.

3.Figure 3 legend: “(c).” but not “c:”

I had changed the c: to (c).

4.Format of the references is not unified

All the references and citations were prepared again according to the MDPI general rules. The final format like the following,

  1. Allan, G.M.; Ellis, J.A. Porcine circoviruses: a review. J Vet Diagn Invest 2000, 12, 3-14, doi:10.1177/104063870001200102.

Reviewer 2 Report

It is well known that both porcine circovirus (PCV) and porcine parvovirus (PPV) cause various diseases and coses economic losses to global swine industry.  Detection of coexistence of PCV and PPV viruses is very interesting and important.  The study is limited by the lack of detection of PCV3 and PCV4, and PPV6 and PPV7 These additional data would greatly increase knowledge about swine viruses.

Major comments:

Lack of criteria of suspected PCV2 associated disease (PCVAD).

From how many farms pigs and sow scome from? What about vaccination (against PCV2 and/or) status? What about helath status? 

For  PCV2, PPV2 and PPV3 detection annealing temperature was 55°C, PPV1 and PPV5 58°C, and 54°C for PPV4 in PCR method. Lack of Table 1 with primer sequences, therefore the change of annealing temperature to 58°C for PCV2, PPV3, PPV2, PPV5  is unclear.

Moreover, why Authors sequenced only PCV2, PPV2, PPV3 and PPV5, but not PPV1 and PPV4? These weakness (lack of PPV$ sequencing) is noted in the manuscript text, but in last sentence of Discusion. If Authors  failed to clone the capsid gene in this study why did not repeted cloning?

In PCR lack of positive and negative controls. What the authors means about special primers?

Prevalence of PCV2 and PPVs on the tissue level

Authors collected five different kinds of tissue samples and 50 cord blood samples and 60 tonsil samples, in results Authors described 520 samples. Please add exact number of tissue samples from pigs. Lack of Table 2 makes data evaluation impossible. Same for  Prevalence of PCV2 and PPVs on the individual pig level and Table 3, and Prevalence rates of double infection on the individual pig level with Table 4. 

If Authors sequenced only 15 15 PCV2-positive tissue samples (pleas add sample types) it should be added to Methods. Same for Six PPV2, four PPV3, two PPV5

All references has incorrect format.

Minor comments:

Abstract - please add tissues samples. Moreover please add more details about Phylogenetic analysis based on capsid genes of PCV2, PPV2, PPV3 and PPV5.

Lack of Tables in tekst

Introduction

Make a space between PCV1,PCV2,PCV3, nucleotides(nt),  genome[1, 2], infection[9], China[10], domestic pigs[14],  in China[15, 16], pig fetuses[17, 18], herds[19], Poland[20], s in 2016[21], PCV2 infection[22-,  vaccine protection[6, 27], Korea [28].This, n pigs[31-35], fection[22, 23, 26, 36], PCV2e[35, 37-39], in 2005[40], pig[41, 42], years[8, 43-45],  field[29, 38, 47], previous study[27], nese strain[51]. 

Delete space between  [7, 8] .  pig[41, 42] .

kidney cells instead kidney cell or use name PK-15 cell line

Collection of samples

Authors collect one tonsil or tonsils? Same lung or lungs etc.

DNA extraction 

3000 x g instead 3000g. Lack of assessment of DNA concentration and purity. Lack of electroforesis details.

Detection of PCV2 and PPV in various samples by PCR 

Please add details about Sequencing method

Author Response

Author's Reply to the Review Report (Reviewer 2)

Comments and Suggestions for Authors

It is well known that both porcine circovirus (PCV) and porcine parvovirus (PPV) cause various diseases and coses economic losses to global swine industry.  Detection of coexistence of PCV and PPV viruses is very interesting and important.  The study is limited by the lack of detection of PCV3 and PCV4, and PPV6 and PPV7 These additional data would greatly increase knowledge about swine viruses.

1.Major comments:

(1)Lack of criteria of suspected PCV2 associated disease (PCVAD).

Clinical signs of PCVAD include wasting with progressive weight loss, lethargy, dark-colored diarrhea, lymphadenopathy, and paleness or jaundice, etc.

The criteria added in the 2.1. collection of samples.

(2)From how many farms pigs and sows come from? What about vaccination (against PCV2 and/or) status? What about health status?

The tissue sample were collected from 82 suspected PCVAD domestic pigs, each of them selected from different farm for confirming the representativeness of the samples.

The cord blood and tonsil samples randomly come from 15 sow farms, having 100,000 sows and 2,000,000 finisher per year. 

The piglets were vaccinated PCV2 inactivated vaccine at 14 and 35 days after birth, and the sows were not vaccinated any kinds of PCV vaccine and without any obvious clinical symptoms.

(3)For PCV2, PPV2 and PPV3 detection annealing temperature was 55°C, PPV1 and PPV5 58°C, and 54°C for PPV4 in PCR method. Lack of Table 1 with primer sequences, therefore the change of annealing temperature to 58°C for PCV2, PPV3, PPV2, PPV5 is unclear.

Table 1 had been added in the manuscript text, in 2.3. detection of PCV2 and PPVs in various samples by PCR.

(4)Moreover, why Authors sequenced only PCV2, PPV2, PPV3 and PPV5, but not PPV1 and PPV4? These weakness (lack of PPVS sequencing) is noted in the manuscript text, but in last sentence of Discusion. If Authors failed to clone the capsid gene in this study why did not repeted cloning?

The author didn’t sequence the PPV1 due to the low prevalence rates of it. The prevalence rates of PPV1 is 2.9%.

Phylogenetic analyses of PPV4 did not done because we failed to clone the capsid gene in this study. The author deleted the last sentence of discussion.

(5)In PCR lack of positive and negative controls. What the authors means about special primers?

Would you mind telling me what sample should be used as positive and negative controls.

I agree with you that specific primers instead special primers.

(6) Prevalence of PCV2 and PPVs on the tissue level

Authors collected five different kinds of tissue samples and 50 cord blood samples and 60 tonsil samples, in results Authors described 520 samples. Please add exact number of tissue samples from pigs.

We collected five different kinds of tissue samples from each pig and in total 82 pigs.

82*5=410

410+50+60=520

(7) Lack of Table 2 makes data evaluation impossible. Same for Prevalence of PCV2 and PPVs on the individual pig level and Table 3, and Prevalence rates of double infection on the individual pig level with Table 4.

All the tables will be submitted as attachment material.

(8) If Authors sequenced only 15 15 PCV2-positive tissue samples (pleas add sample types) it should be added to Methods. Same for Six PPV2, four PPV3, two PPV5

Sample types had been added in the context 2.4, Sequencing and phylogenetic analyses.

(9)All references has incorrect format.

All the references and citations were prepared again according to the MDPI general rules. The final format like the following,

Allan, G.M.; Ellis, J.A. Porcine circoviruses: a review. J Vet Diagn Invest 2000, 12, 3-14, doi:10.1177/104063870001200102.

2.Minor comments:

(1)Abstract - please add tissues samples. Moreover please add more details about Phylogenetic analysis based on capsid genes of PCV2, PPV2, PPV3 and PPV5.

The author had added the list of tissue samples and some details about Phylogenetic analysis.

(2)Lack of Tables in tekst

Sorry, I don’t get it.

(3)Introduction

Make a space between PCV1,PCV2,PCV3, nucleotides(nt),  genome[1, 2], infection[9], China[10], domestic pigs[14],  in China[15, 16], pig fetuses[17, 18], herds[19], Poland[20], s in 2016[21], PCV2 infection[22-,  vaccine protection[6, 27], Korea [28].This, n pigs[31-35], fection[22, 23, 26, 36], PCV2e[35, 37-39], in 2005[40], pig[41, 42], years[8, 43-45],  field[29, 38, 47], previous study[27], nese strain[51].

A space was added in the corresponding position.

(4)Delete space between  [7, 8] .  pig[41, 42] .

The space had been deleted.

(5)kidney cells instead kidney cell or use name PK-15 cell line

Modified according to your advice

(6)Collection of samples

Authors collect one tonsil or tonsils? Same lung or lungs etc.

Just collected one tonsil or lung.

(7)DNA extraction

3000 x g instead 3000g.

Modified according to your advice.

Lack of assessment of DNA concentration and purity. Lack of electroforesis details.

The assessment of DNA concentration and purity was conducted by the TSINGKE Biological Technology (Wuhan, China) before sequence.

(8)Detection of PCV2 and PPV in various samples by PCR

Please add details about Sequencing method

Owing to the sequencing process was conducted by third party firm, The author lack knowledge of it.

Reviewer 3 Report

PCV2 is involved various disease conditions as authors described, especially in a form of co-infection with other pathogens, such as PPV that authors studied. Authors described why PPV was selected to study together with PCV2 at the end of the introduction, but this should be at the top of the introduction.

Overall, English spells and spaces, and grammars were found to be thoroughly corrected in detail, and the indentation is different. Arbitrarily narrow or normal character spacing is observed.

Also, there is not line numbers to be used in review.

The references cited by the authors differ in severity when dealing with two or more pathogens. However, the author is equally studying PCV2 and PPV with equal weight. Then, it is doubtful that the validity of why the study was conducted on these two pathogens is not mentioned.

As grammatical errors as well as English expressions are observed throughout the manuscript, English correction is required.

Author Response

Author's Reply to the Review Report (Reviewer 3)

1.PCV2 is involved various disease conditions as authors described, especially in a form of co-infection with other pathogens, such as PPV that authors studied. Authors described why PPV was selected to study together with PCV2 at the end of the introduction, but this should be at the top of the introduction.

The author had put the content why PPV was selected to study together with PCV2 at the top of the introduction according to your advices.

2.Overall, English spells and spaces, and grammars were found to be thoroughly corrected in detail, and the indentation is different. Arbitrarily narrow or normal character spacing is observed.

The author had adjusted the indentation and the Arbitrarily narrow or normal character spacing.

3.Also, there is not line numbers to be used in review.

The line numbers have been added in the revised manuscript.

4.The references cited by the authors differ in severity when dealing with two or more pathogens. However, the author is equally studying PCV2 and PPV with equal weight. Then, it is doubtful that the validity of why the study was conducted on these two pathogens is not mentioned.

In order to balance the severity of two pathogens, PCV2 and PPVs, the authors had cited new reference to emphasize the severity of PPVs.

5.Comments on the Quality of English Language

As grammatical errors as well as English expressions are observed throughout the manuscript, English correction is required.

Firstly, we had checked grammatical errors as well as English expressions by ourselves. Secondly, If the revised manuscript was proceeded to the next round of review by the journal, we will send the manuscript to MDPI Author Service for further revise to meet the highest academic standards before publication.
